# Predicting a diagnosis of ankylosing spondylitis using primary care health records– A machine learning approach

**Jonathan Kennedy**[1]*, **Natasha Kennedy**[1], **Roxanne Cooksey**[1], **Ernest Choy**[2], **Stefan Siebert**[3], **Muhammad Rahman**[1], **Sinead Brophy**[1]

1 Data Science Building, Swansea University, Wales, United Kingdom, 2 CREATE Centre, Section of Rheumatology, Division of Infection and Immunity, School of Medicine, Cardiff University, Cardiff, United Kingdom, 3 Institute of Infection Immunity & Inflammation, University of Glasgow, Glasgow, United Kingdom

* j.i.kennedy@swansea.ac.uk

**Data Availability Statement:** The data is owned by the SAIL Databank (https://saildatabank.com/) and all datasets are available from them upon request and appropriate information governance. The

## Abstract

Ankylosing spondylitis is the second most common cause of inflammatory arthritis. However, a successful diagnosis can take a decade to confirm from symptom onset (via x-rays). The aim of this study was to use machine learning methods to develop a profile of the characteristics of people who are likely to be given a diagnosis of AS in future. The Secure Anonymised Information Linkage databank was used. Patients with ankylosing spondylitis were identified using their routine data and matched with controls who had no record of a diagnosis of ankylosing spondylitis or axial spondyloarthritis. Data was analysed separately for men and women. The model was developed using feature/variable selection and principal component analysis to develop decision trees. The decision tree with the highest average F value was selected and validated with a test dataset. The model for men indicated that lower back pain, uveitis, and NSAID use under age 20 is associated with AS development. The model for women showed an older age of symptom presentation compared to men with back pain and multiple pain relief medications. The models showed good prediction (positive predictive value 70%-80%) in test data but in the general population where prevalence is very low (0.09% of the population in this dataset) the positive predictive value would be very low (0.33%-0.25%). Machine learning can be used to help profile and understand the characteristics of people who will develop AS, and in test datasets with artificially high prevalence, will perform well. However, when applied to a general population with low prevalence rates, such as that in primary care, the positive predictive value for even the best model would be 1.4%. Multiple models may be needed to narrow down the population over time to improve the predictive value and therefore reduce the time to diagnosis of ankylosing spondylitis.

## Introduction

Ankylosing Spondylitis (AS) affects 1 in 400 [1] people and is the second most common cause of inflammatory arthritis after rheumatoid arthritis (RA). AS is characterized by spinal (axial)

authors had no special access privileges to the data others would not have.

**Funding:** This work was supported by UCB Pharma, Health Data Research UK, and the infrastructure support of the National Centre for Population Health and Wellbeing and the SAIL Databank. The funders had no input in to the study design, analysis or interpretation or writing up of the work.

**Competing interests:** JK, SB reports grants from UCB Pharma, during the conduct of the study; and involved in (but not directly funded by) grants from Biogen, Sanofi and Novartis. RC reports grants from Pfizer, during the conduct of the study;. EC reports grants from UCB, during the conduct of the study; grants and personal fees from Pfizer, grants and personal fees from UCB, grants from BioCancer, grants from Biogen, grants and personal fees from Novartis, grants and personal fees from Roche, personal fees from Amgen, personal fees from Chugai Pharma, personal fees from Eli Lilly, grants and personal fees from Sanofi, personal fees from Abbvie, personal fees from Janssen, personal fees from Gilead, personal fees from Bristol Myer Squibbs, outside the submitted work; In addition, Dr. Choy has a patent null pending. SS reports grants and personal fees from AbbVie, grants and personal fees from UCB, grants and personal fees from Novartis, grants and personal fees from Janssen, grants and personal fees from Pfizer, grants from Bristol Myers Squibb, from Boehringer-Ingelheim, outside the submitted work; This does not alter our adherence to PLOS ONE policies on sharing data and materials.

**Abbreviations:** ALF, Anonymised Linking Field; AS, ankylosing spondylitis; ASAS, Assessment of SpondyloArthritis Society; CRP, C-reactive protein; MRI, magnetic resonance imaging; NSAID, Non-steroidal anti-inflammatory drugs; SAIL, Secure Anonymised Information Linkage; SpA, spondyloarthritis, HLA-B27, GP, MRI.

inflammation; however, the presentation of the disease differs between men and women, with women reported to have less structural radiographic changes and higher erythrocyte sedimentation rate (ESR) (indicating inflammation) compared to men [2]. The introduction of effective treatment options for AS, such as tumour necrosis factor (TNF) inhibitors [3, 4], means it has become imperative to identify the disease as early as possible to treat patients early in the course of the disease. However, as chronic low back pain is one the most common presenting symptoms in primary care, it is difficult to identify these patients among the far larger pool of people with non-specific chronic low back pain. New advances in imaging, especially in magnetic resonance imaging (MRI), has allowed the earlier recognition of inflammatory change before the irreversible radiographic changes that are used to diagnose AS. However, the early inflammatory changes also detect people who never go on to develop AS or any radiographic changes. The Assessment of SpondyloArthritis Society (ASAS) classification criteria [5] recognises this and patients can fulfil a classification of axial spondyloarthritis (SpA), which incorporates both AS and non-radiographic axial SpA [6]. The ASAS criteria utilise x-rays, MRI and/or clinical and laboratory features to classify patients as having axial SpA with a sensitivity of 68–87%, and specificity of 62–95% against rheumatologist diagnosis [7, 8]; at follow-up a positive predictive value of 93.3% was obtained for the patient being diagnosed with axial SpA at 3–5 years. However, only 5% of patients with non-radiographic axial SpA will progress to AS at 4 years follow-up, indicating a 5% positive predictive value for identifying AS [6]. The positive predictive value increases to 12% if elevated C-Reactive Protein (CRP) levels and active sacroiliitis on MRI are present [6]. This implies that identifying axial SpA alone does not help to identify early or pre-AS and so is not a strong predictor of progression to AS. While many patients with non-radiographic axial SpA will require treatment based on their level of symptoms, it is important to identify early those who will go on to develop radiographic progression (i.e. AS) as these changes are irreversible and associated with greater functional limitation. However, predicting those who develop radiographic damage (AS) using traditional methods remains difficult due to the requirement for longitudinal data.

This study aimed to undertake a new approach in predicting the development of AS. The expansion of Big Data and high performance computing enables routinely-collected patient records to be linked, thus, enabling the patient journey to be tracked through the health care system over long periods of time [9]. Novel machine learning methods can potentially identify patterns and clusters of terms/data, such as diagnosis, procedures, and medications, which are observed more frequently in people who have a subsequent diagnosis of AS, compared to people who do not receive a diagnosis of AS. However, the underlying theories in data-driven methods for prediction are very different from those of traditional causal modelling. Predictive modelling looks for associations with the outcome but is not looking to minimise confounders or look for causal pathways. This means factors which are predictive of an outcome may bear little relation to factors that are on the causal pathway for that outcome (for example, multiple blood tests for liver function may be predictive of a diagnosis of arthritis due to association with medication, but are not related to the causal pathway for arthritis). This study uses machine-learning to identify diagnostic codes, medications, procedures and administrative data of patients who are given a future diagnosis of AS compared to controls and examines the patient journey through the National Health Service (NHS) for people with AS.

## Materials and methods

### Dataset

The Secure Anonymised Information Linkage (SAIL) databank is a national (Wales, United Kingdom) data repository which allows person-based data linkage across datasets. This

databank includes Welsh primary care (general practitioner, GP) data, hospital in- and out-patient records, as well as mortality data collected by the Office of National Statistics (ONS). SAIL comprises over a billion anonymised records. It employs a split-file approach to ensure anonymisation and overcome issues of confidentiality and disclosure in health-related data warehousing. Demographic data is sent to a partner organisation, NHS Wales Informatics Service, where identifiable information is removed; clinical data are sent directly to the SAIL Databank and each individual is assigned an encrypted Anonymised Linking Field (ALF). The ALF is utilised to link anonymised individuals across datasets, facilitating longitudinal analysis of an individual's journey through multiple health, education and social datasets [10].

Data collected by GPs is captured via Read Codes (5-digit codes related to diagnosis, medication, and process of care codes) [10]. The secondary care dataset utilised in this study is from the clinical system, Cellma by Riomed [9], employed in the rheumatology departments of local hospital heath boards; Swansea Bay University Health Board (SBU—Swansea, Neath and Bridgend areas), and Cardiff and Vale University Health Board (CVUHB). This commercial system applies SNOMED-CT to code diagnosis and medications [9], as well as recording clinical data entered by rheumatologists at the point of clinical contact. The Cellma systems were available in the SBU region from March 2009 until October 2012, and in Cardiff from October 2013 to July 2014. Hospital in- and out-patient data are collected in the Patient Episode Database for Wales, which contains clinical information regarding patients' hospital admissions, discharges, diagnoses and operations utilising the International Classification of Diseases (ICD-10) clinical coding system. The ONS mortality dataset contains demographic data, place of death and underlying cause of death (also ICD-10).

## Ethical approval

Data held in the SAIL databank are anonymised, consequently, no ethical approval is required. All data contained in SAIL has permission from the relevant Caldicott Guardian or Data Protection Officer. SAIL-related projects are required to obtain Information Governance Review Panel (IGRP) approval and this study had governance approval.

## Patient selection and definition

Early pilot investigation demonstrated differences in the presentation of AS for men compared to women, for example women had less back pain codes but more pain medication codes. For this reason, the data was stratified so that analysis was run separately for males and females. The raw data was selected and then cleaned before matching AS patients with the general population.

Patients with AS were selected as cases and people from the general population without AS were selected as controls. A person was selected as having AS if they had a diagnosis of AS in the GP data (Read code N100), hospital admissions (ICD10 code M45) or Rheumatology clinical system. The controls from the general population were selected as those with no evidence of AS or axial SpA within their records (see Fig 1). People with AS were included if they were diagnosed between the ages of 15 and 35 years (this age was chosen in line with the NICE pathway [11], diagnosed after the year 2000 and were registered with a Welsh GP a minimum of 3 years before the AS diagnosis date (some had 7+ years of data but the minimum dataset was 3 years). These criteria were chosen to ensure there was sufficient high quality data to map the patient journey prior to the diagnosis of AS. These restrictions meant that of 5751 AS male patients identified (see S1 Fig), 543 (9.4%) males with full patient records were used in developing the model (380 in model creation, 163 in testing the prediction of the model) and 250 (9.2%) of the 2702 females with AS (see S2 Fig) were used in the model (175 in developing the

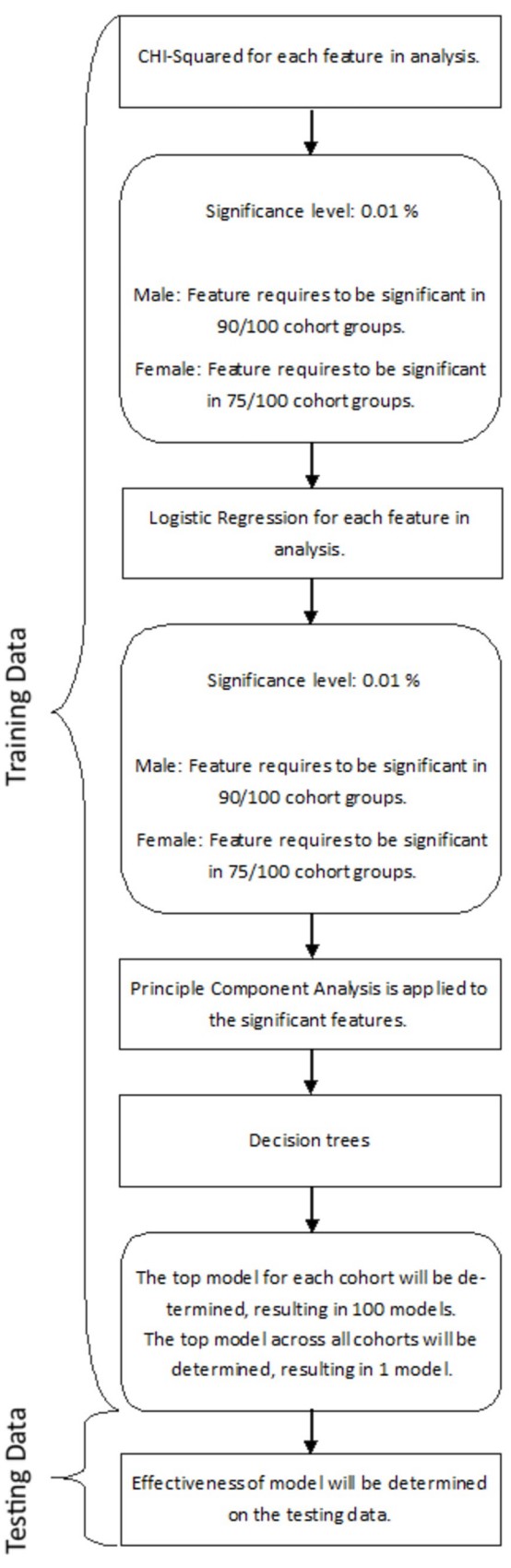

**Fig 1. Overview of analysis.**

model, 75 in testing the prediction of the model). In this study, 10% (n = 543/5751) of patients were used to develop the model and the remaining patients with a diagnosis between the ages of 15 to 35 years (n = 1559/5751, 27% of total) were used to examine the generalizability of the model. This means 63% of the data was not used in the analysis as these patients were diagnosed with AS before the age of 15 or after the age of 35.

The date of diagnosis of AS was recorded as the first mention of AS in GP, hospital admission or rheumatology records. An earlier 'suspected AS' date was recorded if patients were prescribed an anti-TNF inhibitor before the first mention of AS, or if the patient had a record for an HLA-B27 test or a diagnosis of "spondyloarthropathy" and subsequent diagnosis of "AS". The presence of HLA-B27 can also be associated with other conditions such as uveitis. Finally, the AS patients were matched 1 to 100 with controls in the general population. Matches were on diagnosis date (for the AS case) and an equivalent time date selected at random (for controls), sex and week of birth (within 6 months). All selected general population controls were required to have 3 years of GP data before and after the random date to limit the chance of misclassification of AS. Therefore, the patients have to survive at least 6 years to be in the control dataset and they need to survive at least 3 years to be in the AS dataset. People who had less than 6 years of data in the GP system could not be selected as controls.

## Feature construction for analysis

All healthcare data is recorded at set points in time (e.g. date of GP appointment) even though it may reflect ongoing treatment and care. In addition, the high complexity of the data results in very long data processing times. Both these issues can be overcome by running the analysis at slices in time separately, e.g. an unspecific back pain code present between age 15 and 20 AND a pelvic pain codes present between age 25 and 30. This enables a time step pathway of codes which become relevant as the disease develops. This work can identify the difference between codes found before, after and overlapping the incident date of diagnosis.

In routine data some important codes, such as diagnosis, may have only one occurrence, while some common codes, such as procedures (e.g. blood pressure check), may be recorded very frequently. This means procedure codes will end up having a higher weighting in any signal detection compared to diagnosis. To minimise the influence of certain codes being highly abundant (e.g. prescriptions) and other codes only appearing once or twice (e.g. diagnosis), the variables are expressed in a binary form; e.g. any code for diagnosis of iritis, any mention of a prescription of NSAID between age 15 to 20. In this example, the frequency of a diagnosis code of iritis may be 1 and the frequency of NSAID code may be 12 prescriptions but both are recorded in the data set as a binary "1" i.e. present.

Another issue with routine data is that coding can go to very specific details and levels, such as one code for suspension preparation and another code for tablet preparation of the same drug. For this reason the analysis was repeated to examine all levels of codes, with every level examined separately. Additionally, codes are aggregated so that concepts can be included in the features (e.g. a "pain" code flag may aggregate all the specific pain codes (pain in left thumb and pain in lumbar spine) into one combined "pain present" code. Finally, the laboratory test codes used in this analysis were simply "had test", but the test result was not analysed. The ability to interpret and use the test results (using the laboratory dataset) was beyond the scope of this work.

All the codes were labelled so they could be analysed to distinguish the origin of the code (i.e. hospital or GP or specialist clinical dataset), number of characters in the code (which gives the level of specificity), and time (e.g. between the ages of 15 and 20 years).

## Statistical analysis

### Feature/Variable selection

Participants were split into a training (70%) and a testing (30%) group for the male (S1 Fig) and female (S2 Fig) cohorts. The different stages of the analysis were undertaken 100 times with the same affected AS case patient compared with 100 different matched controls taken from the general population. Hence, the Chi-squared tests were run in batches of (a) AS cases (n = 380 (male), n = 175 (female)) vs. general population control 1 (n = 380 (male), n = 175 (female)), (b) AS cases (n = 380 (male), n = 175 (female)) vs. general population control 2 (n = 380 (male), n = 175 (female)), this was repeated for all 100 controls. The intention of this method was to compare the AS cohort to a representative sampling of individuals with the general population. Therefore, one AS patient characteristic was represented in 100 tests; 79300 tests were run for all 793 AS patients and 100 controls each for the male and female cohorts. Then, the first stage of the analysis (see Fig 1) utilised Chi-squared tests to identify which codes were more frequent in the AS or the control at each time point (e.g. every year). Thus, 100 Chi-squared tests were run for each of the 3 years preceding the diagnosis date (700 chi-squared tests), to identify significant differences between the frequencies of presence and absence of selected codes in AS compared to general population patients. A code was taken as being associated with AS (or not associated with AS but associated with general population) if it achieved a significance level 1% ($p < 0.01$) in 90% of the chi-squared tests for males and in 75% of chi-square test for females (as we had a lower sample size and so lower power in females), for the time point selected. The codes that reached this level of significance were taken forward to be included in logistic regression tests. To remain in the logistic regression model, a code needed to have 1% significance ($p < 0.01$). This list of significant variables was then transformed utilising principal component analysis to better understand the structure and relationship of the features to each other.

### Model development

The principal components were then utilised to create a decision tree to better understand sub-groups identified within the analysis. An individual decision tree was identified for each of the 100 cohorts under investigation. To select the most predictive model, the prediction of the model was compared with the training dataset raw data and the models with the highest F value are selected. The models were compared with the 100 match pair groups (e.g. AS vs. general population control 1, AS vs. general population control 2 etc.). This resulted in 100 training models being identified. The model selected is the one with the highest average F value across the 100 runs.

### Model validation

The selected model is applied to the testing dataset and its effectiveness (e.g. sensitivity, specificity, Positive Predictive Value, Negative Predictive Value) is recorded.

### Model generalisablity

The model was then used to predict AS development in the general population to identify those who went on to develop AS but had not met the minimum level of data inclusion criteria to be included in the model development.

## Results

### Predictive features

The model developed for the male and female cohorts assesses the codes at different ages and weights them to give an overall total score for the individual (see S1 and S2 Tables). For example, an individual who is male aged 20 to 25, who hasan ESR test, diagnosis of musculoskeletal or connective tissue diseases, prescribed NSAIDs, with many laboratory procedures and has a previous history of vertebral column syndromes at age 15 to 20, would score highly for suspected AS. In general, the predictive features identified included;

- Musculoskeletal diagnosis.

- Pain codes.

- Pain medications

- Blood tests.

- X-rays of the spine.

- Uveitis.

Codes associated with a musculoskeletal diagnosis at a younger age (e.g. age 15–20) were associated with a higher chance of a future diagnosis of AS. In temporal order these early conditions were then followed by investigations (e.g. laboratory tests such as lymphocyte count, ESR test) at age 20 to 25 before being associated with the development of AS. In contrast, laboratory procedures between 30 and 35 years were weighted by the model as less likely to be associated with AS. This aspect might be partially associated with the selection cut off criteria for the cohort creation.

Women with AS had a greater number of additional musculoskeletal diagnoses and increased numbers of diagnoses from hospitals (S2 Table); suggesting that women have a more complicated path to being diagnosed with AS, with multiple tests, diagnoses, and referrals, compared to men with AS.

The codes encompassing pain and pain drugs are also significant features. The pain codes were predominantly general pain and specific back pain without radiation. Within the model, a young male or female attending the GP with prolonged pain and given pain relief medication is weighted as a higher chance of AS.

Numerous blood tests were identified as predictors for the AS model; these are utilised as an investigative tool to ascertain causes of symptoms experienced by the patient. In clinical practice, these tests are undertaken in a batch process to identify causes, as well as markers of other conditions, thus reducing the possible diagnosis. The codes associated with rheumatoid factor (a test to identify if the patient might have RA), are present for males and females; however, women are more likely to be given the test earlier than men. This is likely due to AS symptoms in females being suspected as a result of other conditions, such as RA.

The males with AS in the study had x-rays of the spine between the ages 15 to 20 while women generally did not have x-rays until aged 25 and 30. Instead, women were more likely to have codes for sciatica, back pain, and prescriptions for pain relief between 20 and 25 years old, and older. This may imply that there is an increased reluctance to consider a diagnosis of AS or to perform spinal or pelvic x-rays for young female patients compared to their male counterparts.

Uveitis is a significant extra-articular manifestation in people with AS [12]. However, this only appeared as a feature in the male population in our cohort using this method. There was a

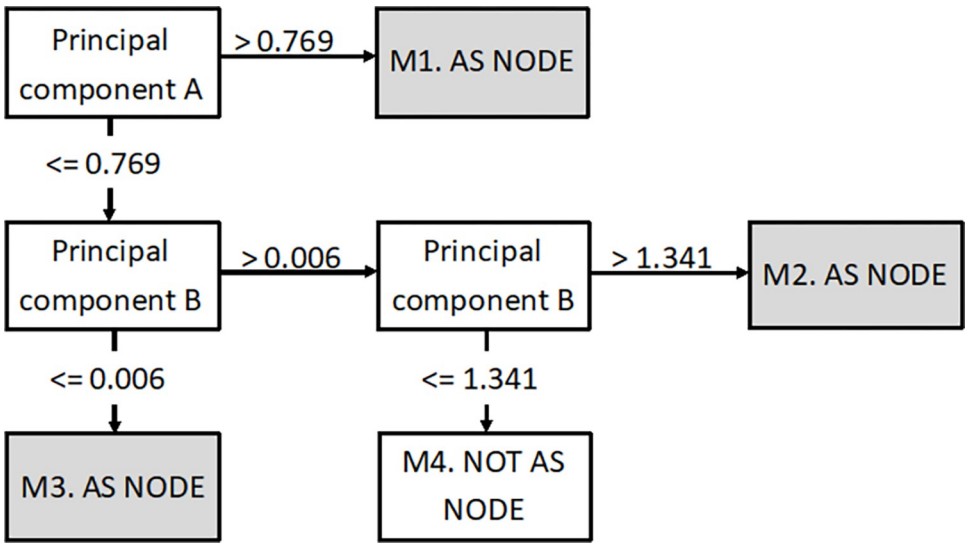

**Fig 2. Combined results of machine learning (male).**

smaller female cohort for analysis, which could have meant the study was underpowered to detect uveitis as a predictor in women.

## Decision tree for male cohort

The principal components analyses were models of features/variables that were weighted to give a final score for an individual. These models were combined into decision trees. The tree for males (Fig 2) shows that if their principal component A score is above 0.769, they would have a high chance of an AS diagnosis (see Fig 2.M1) in the future (this model will identify people who have had symptoms from teenage years). If their score is lower than 0.769 then they were considered using principal components model B, where a very low score (less than 0.006 (see Fig 2.M3)) or a very high score (more than 1.341 (see Fig 2.M2)) would indicate a high chance of AS. A very low score in principal component B would be someone with multiple tests when they are aged 25 to 30 including rheumatoid factor, radiographs, x-rays, and taking pain killer drugs. A very high score would be someone taking NSAIDs when young (prescribed diclofenac/ ibuprofen when aged 15 to 20). However, a score in principal component model B between 0.006 and 1.341 (see Fig 2.M4) would mean they are less likely to have AS (e.g. blood tests were when they were 30 and they have no history of problems when they were 15 to 20). The model validation (i.e. predictive value in the 30% test data split) is presented in Tables 1–3 showing a positive predictive value of 76.69%.

## Decision tree for female cohort

The decision tree generated for the female cohort was more complex and had six possible outcomes (see Fig 3), four with higher chances of developing AS and two associated with lower

**Table 1. Training and testing results (male).**

|  | Sensitivity (%) | Specificity (%) | Positive Predictive Value (%) | Negative Predictive Value (%) | Accuracy (%) | F Value (%) |
|---|---|---|---|---|---|---|
| Training | 59.74 | 81.76 | 76.65 | 67.00 | 70.75 | 67.14 |
| Testing | 60.74 | 81.45 | 76.69 | 67.46 | 71.09 | 67.77 |
| Validation | 21.43 | 82.25 | 0.15 | 99.88 | 82.17 | 0.30 |

**Table 2. Male all population.**

| | | PREDICTED | | |
|---|---|---|---|---|
| | | AS | NON_AS | |
| OBSERVED | AS | 765 | 1337 | 36.39% |
| | NON_AS | 229815 | 1064605 | 82.25% |
| | | 0.33% | 99.87% | 82.17% |

chances of developing AS. In Fig 3.F1, the node is associated with a high value (>2.138) using principal component C and is associated with a higher probability of AS. This node is related to codes within the years 20 to 25 for back pain, NSAIDs, and blood tests. This is also related to a group of codes between 25 to 30 years age, including; diclofenac sodium, radiology, vertebral column syndromes, and other back disorders. This implies that this group of patients present with back pain and are being investigated at 20 to 25 years, but x-rays and diagnosis of AS might not be made until they are 25 to 30 years old.

Female patients that correspond to Fig 3.F2 have features at 25 to 30 years for backache symptoms, laboratory tests, and diclofenac sodium. This is then followed by NSAIDs and musculoskeletal diagnosis at 30 to 35 years old. Like node F there are additional diagnosis of musculoskeletal conditions prior to AS diagnosis and it takes several years for a successful diagnosis, albeit at a later age than for Fig 3.F1.

Fig 3.F2 is also predictive of AS and is characterised by features across three age groups. The first group is related to joint pain between 20 and 25 years old. This is followed by x-rays of the spine at 25 to 30 years old and then diclofenac sodium at 30 and 35 years old. This grouping also points to a prolonged time between initial symptoms and a successful diagnosis of AS. Fig 3.F3 is also predictive of AS and has codes at a young age followed by a very long time to successful diagnosis. There are codes for arthropathies at age 15 to 20 years old. This is followed by features at 25 to 30 years old for sciatica and co-dydramol tablets.

The Fig 3.F4 node represents a non-AS prediction and is associated with a lower score using principal component C which suggests having laboratory tests (serum CRP level, plasma viscosity, ESR, enzymes/specific proteins) undertaken at age 15 to 20 years old. Node Fig 3.F6 is the other non-AS group and the patients within it were characterised by not being similar to the other groups.

Taken together this model suggests multiple pathways before obtaining a diagnosis for women with AS. The positive predictive value of the model for female patients was 78.3% in the testing dataset (see Tables 4–6).

## ROC curves

The following ROC curves (Figs 4 and 5) show the results of the male and female groups under investigation. The area under the curve for the male cohort is 0.870519215 and the female cohort has an area under the curve of 0.863183728.

**Table 3. Male excluding training and testing data.**

| | | PREDICTED | | |
|---|---|---|---|---|
| | | AS | NON_AS | |
| OBSERVED | AS | 339 | 1243 | 21.43% |
| | NON_AS | 229815 | 1064605 | 82.25% |
| | | 0.15% | 99.88% | 82.17% |

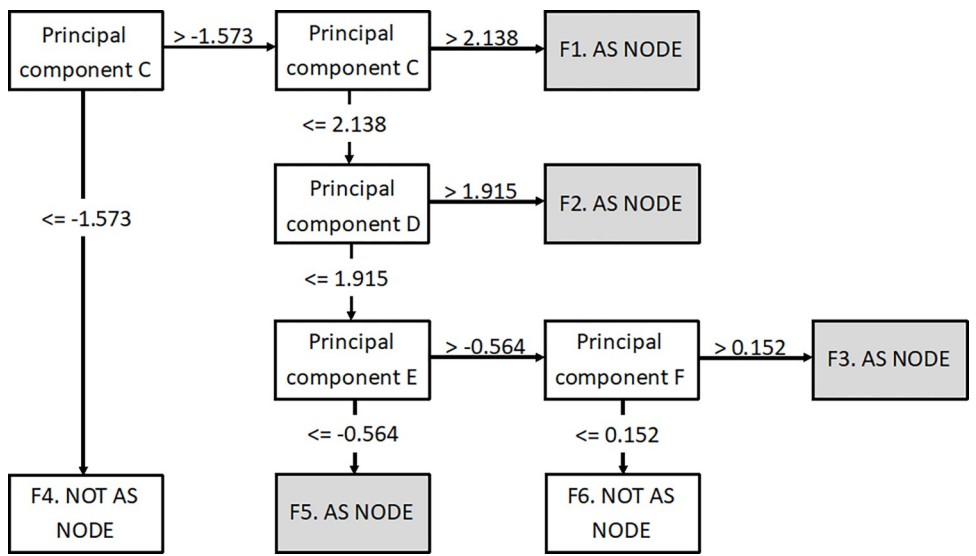

**Fig 3. Combined results of machine learning (female).**

**Table 4. Training and testing results (female).**

|  | Sensitivity (%) | Specificity (%) | Positive Predictive Value (%) | Negative Predictive Value (%) | Accuracy (%) | F Value (%) |
|---|---|---|---|---|---|---|
| Training | 48.57 | 88.43 | 80.77 | 63.23 | 68.50 | 60.66 |
| Testing | 45.33 | 87.44 | 78.30 | 61.53 | 66.39 | 57.42 |
| Validation | 52.72 | 87.99 | 0.25 | 99.97 | 87.97 | 0.50 |

**Table 5. Female all population.**

|  |  | PREDICTED |  |  |
|---|---|---|---|---|
|  |  | AS | NON_AS |  |
| OBSERVED | AS | 397 | 356 | 52.72% |
|  | NON_AS | 158025 | 1158198 | 87.99% |
|  |  | 0.25% | 99.97% | 87.97% |

## Model generalizability

The model was the applied to the unused individuals who had been removed though the selection criteria (see S1 and S2 Figs). These AS patients were grouped with all the individuals in the GP system that had no AS within their history and have 1 or more days of data between the age of 15 and 35 (i.e. the model was validated on the general GP population in Wales with the individual who were used to develop the model removed, therefore giving a lower than normal prevalence of AS in the validation dataset).

The results for the model generalizability are shown in Tables 1–3 (male) and Tables 4–6 (female). The specificity achieved in this was comparable to the training/testing cohorts (82% (male) 88% (female), while the sensitivity was lower (21% in males and 52% in females). This is likely due to the general population cohort having incomplete coverage of data compared to the other cohorts. The positive predictive value is considerably lower in the general population cohort dataset as this is dependent on the prevalence of the condition in the population. As a

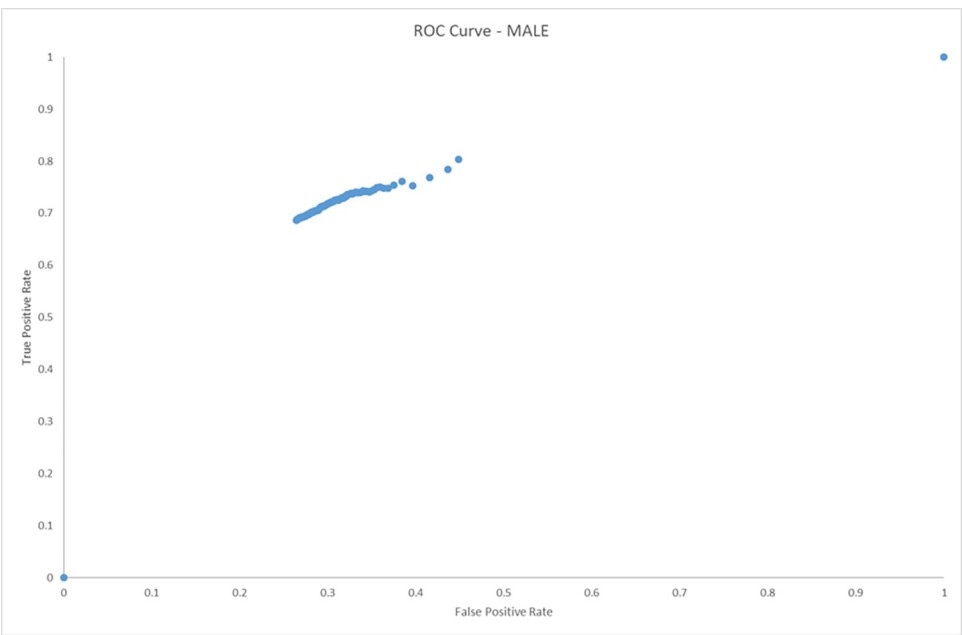

**Fig 4. ROC curve (male only).** AUC (area under curve) = 0.870519215.

result of the way it was defined, in the general population cohort dataset 0.09% of the population had AS compared to 1% in the (1 to 100 controls) in the test dataset. There were some codes that were only found in the female analysis (see S3 Table) such as sciatica and rheumatoid factor at a younger age, and some codes only found in the male analysis, such as uveitis.

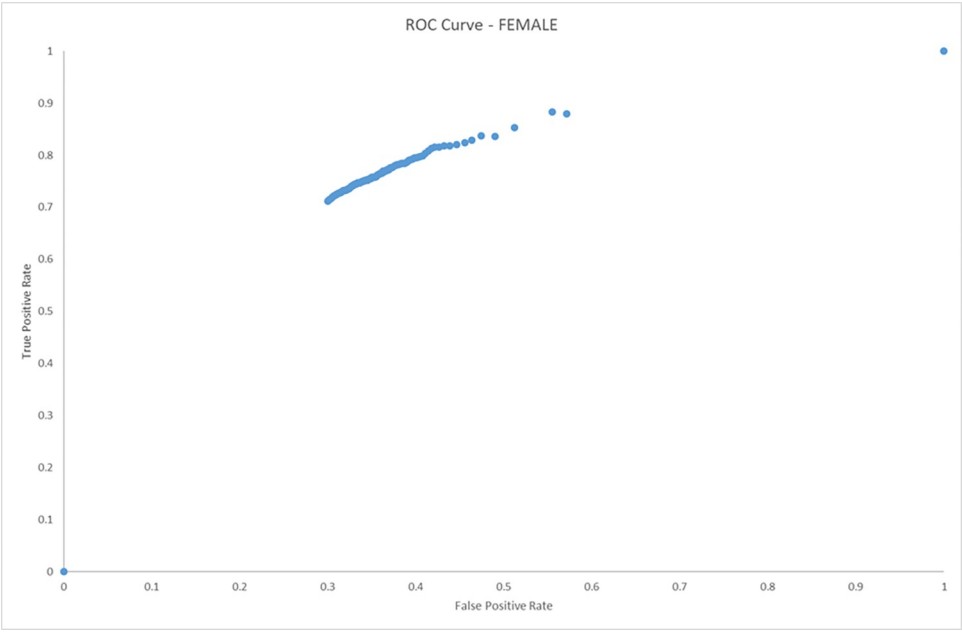

**Fig 5. ROC curve (female only).** AUC (area under curve) = 0.863183728.

**Table 6. Female excluding training and testing data.**

|  |  | PREDICTED | |  |
|---|---|---|---|---|
|  |  | AS | NON_AS |  |
| OBSERVED | AS | 223 | 280 | 44.33% |
|  | NON_AS | 158025 | 1158198 | 87.99% |
|  |  | 0.14% | 99.98% | 87.98% |

## "Perfect" model vs. developed model

To be able to determine the reliability and accuracy of a machine learning model on general population data, the mode of assessment needs to be put in context. The model will be assessed against a "perfect" model to be able to determine if the model can be considered successful. If we make the assumption that a "perfect" model would have 90% sensitivity and specificity as higher percentages than this would likely be attributed to overfitting of the model. The most important comparison point is the positive predictive value, as this will determine the ratio of correctly characterised patients. The equation for positive predictive value is;

$$Percentage\ of\ AS = \frac{(Total\ observed\ AS + incorrectly\ predicted\ AS)}{Total\ population}$$

The prevalence in the datasets, using the "perfect" model for males (see Table 7) would have obtained a positive predictive value of 1.44%, (based on prevalence rates seen in the validation dataset) compared to 0.33% which was obtained with our model (see Table 5). The percentage of AS cases identified for the "perfect" model was 10.1% (i.e. a high rate of false positives), while in our model this was 17.9%. Therefore, these results indicate that both models err in identifying too many suspected AS cases (which are not truly AS). This is likely due to the inclusion of commonly used codes in the GP records, such as pain codes.

The female population obtained a positive predictive value of 0.51% for the "perfect" model (see Table 8), while the positive predictive value for our model was 0.25% (Table 5). The percentage of the "perfect" model identified 10% of the dataset as AS compared to 12% for the model created. These features identify the female model as being closer to the "perfect" model than the male model. This implies that the female model is more accurate than the male model at identifying patients with AS.

**Table 7. Perfect model output (male).**

|  |  | PREDICTED | |  |
|---|---|---|---|---|
|  |  | AS | NON_AS |  |
| OBSERVED | AS | 1891.8 | 210.2 | 90.00% |
|  | NON_AS | 129442 | 1164978 | 90.00% |
|  |  | 1.44% | 99.98% | 90.00% |

**Table 8. Perfect model output (female).**

|  |  | PREDICTED | |  |
|---|---|---|---|---|
|  |  | AS | NON_AS |  |
| OBSERVED | AS | 677.7 | 75.3 | 90.00% |
|  | NON_AS | 131622.3 | 1184601 | 90.00% |
|  |  | 0.51% | 99.99% | 90.00% |

## Discussion

AS is an important inflammatory musculoskeletal condition with effective therapy available, especially if identified early. However, there is no single diagnostic test for AS and symptoms over-lap with many other more common conditions, especially non-specific low back pain. Machine learning offers a new way of approaching the early identification of people with AS and to address the delayed diagnosis that characterises this condition, especially for women [13]. The findings from this study showed that although machine learning methods can achieve very high positive predictive value within test datasets, it must be recognised that with low prevalence of AS in the general population, the positive predictive value in general and primary care clinical settings will always be very low.

The model to predict AS development performed well with 76.69%-78.3% positive predictive value within the test sets, which is equivalent to other publications in this field and equivalent to that predicting AxS [13]. The developed model performed better for women with AS than for men. In general it pointed to symptom presentation for women in the age range of 20–30 (which differs from men who present with back pain in their teens). The women present with pain, unspecified back disorders, musculoskeletal connective tissue disorders as coding items.

The 'perfect' model in females would have had positive predictive values of 1.4% and our model achieved a 0.25%, this suggests that not all the predictive variables have been captured in the dataset. The next stage would be to take the profiles of the decision trees forward so that we can identify people who fit the profile but are not diagnosed, and seek additional data about these patients from both medical records (e.g. results of laboratory tests such as elevated CRP) and to incorporate information directly from patients via an online patient reported measures questionnaire to capture more granularity and current symptoms (e.g. duration of pain, stiffness and pain worse in morning, pain improved with movement, buttock pain, hot burning joints, family history of SpA). This could help to develop a clinical tool using data driven methods that might be used online for GPs and/or patients to quantify the probability of AS in order to prioritise and target HLA-B27 testing and MRI scans, and suggest possible referral to a rheumatologist. However, further research would be needed to examine the cost-effectiveness of this strategy.

Limitations: machine-learning models are very dependent on data quality and accuracy. The results of laboratory tests were not used in this model as they were too complex to interpret in the time available. However, the addition of test results is another step that can be taken by future research to improve the prediction of the model. The use of machine learning in the health care field is still in its infancy and this work can be used as a step to improve this process and time to diagnosis, especially for women with AS who have not well studied or recognised.

The model was built using only 9% of all available cases as inclusion criteria selected only those with a diagnosis in the last 18 years and who have at least 3 years of data available for pre-diagnosis profiles. This is a very specific cohort of patients, as both the cases and controls needed to be alive for the study period and needed to stay resident in Wales and given the majority of the patients are young, this means those who are mobile (e.g. moved outside Wales for employment or education) are missing from the dataset. This means the model may be more likely to be based on the people who are more ill and not able to seek work outside of Wales. However, the same could be said for the control population who needed to have 6 years of continuous data (to ensure they did not have AS) to be eligible, so would also be likely to be a stable and relatively non-mobile population. Additionally, there can be a significant delay between onset of conditions and confirmed diagnosis, this can cause misclassification. The impact of this has been limited in the design of the dataset, however misclassification is still possible.

The model identifies a large variety of codes that are significant for identifying AS, however many other codes that would improve the positive predictive value have too low numbers to be useful for the machine learning model as they would be to specific. A further layer of detail could be applied with a more traditional methodology, such as including the results from lab tests, allowing stratification at a number of levels.

In clinical practice, one would want to identify all patients with axial SpA. However, our model utilised historical data and used only codes for AS (i.e. radiographic axial SpA), so did not include the more recent term non-radiographic axial SpA which is not incorporated in GP codes yet (but is in the rheumatology database). Including the full spectrum of axial SpA would incorporate greater heterogeneity but also increase the likelihood of early diagnosis (and therefore earlier treatment) and potentially increase the predictive value due to higher prevalence in the general population.

The models presented here identified the pathways through the health system experienced by people prior to being given the diagnosis of AS. The profiles indicate many codes for pain, drugs to relieve pain, related conditions, and testing for conditions. These features indicate a pattern of repeated interactions with healthcare providers prior to diagnosis. Patients with features fitting a profile of suspected AS could be flagged for consideration of an HLA-B27 test; if positive, a MRI, and possible referral to a rheumatologist for assessment with recommendation for appropriate exercises, smoking cessation, dietary advice and awareness of the increased possibility of associated eye, bowel and skin symptoms [14].

The AS male population can be characterised into 3 main groups;

- Group 1- age under 20, have lower back pain, on medication (Diclofenac Sodium), may develop uveitis in their 20's.

- Group 2- Ages 20 to 25, back disorders, NSAIDs, blood tests

- Group 3—Aged 25 to 30, numerous blood tests, various musculoskeletal codes, back pain and radiology.

The AS female population can be characterised into these separate groups;

- Group 1- identifies individuals that have been identified with arthropathies at 15 to 20 years followed by pain in lumbar spine, co-dydramol and sciatica at 25 to 30 years.

- Group 2—Aged 20 to 30; likely to be prescribed multiple pain medications, be diagnosed with musculoskeletal conditions and had various blood tests. This would evolve by age 25 to 30 years to more back pain codes, vertebral column syndrome, and additional pain relief

- Group 3- identifies a pathway of pain/arthropathies at 20 to 25 years, followed by x-ray to attempt to identify the condition (25 to 30 years) and then Diclofenac Sodium at 30 to 35 years.

- Group 4—Aged 25 to 30 years; This group includes backache symptoms and multiple blood tests.

The techniques presented in this paper created a simpler profile for male when compared to female participants, even though the female population is smaller. This might be due to AS developing in the teenage to early twenties, and males and females having a different healthcare usage at this age. With young men tending to have a much lower usage than females so will likely have a simpler profile in their healthcare record. An additional reason might be that the symptoms expressed in a male would be identified as AS quicker, compared to females who might investigate other conditions before AS, such as rheumatoid arthritis.

Additional criteria from the NICE pathway [11] which were not included in the dataset (as they require patient report) include waking during the second half of the night due to symptoms, pain improved with movement and a first degree relative with spondyloarthritis.

Further work could examine the cost effectiveness of using profiling to identify people for HLA-B27 and/or MRI testing and if positive referral to a rheumatologist. Similarly, excluding AS early may accrue healthcare savings by avoiding further unnecessary investigations and consultations.

## Conclusions

This study indicates that machine learning has the potential to help identify people with AS and better understand their diagnostic journeys through the health system, but we need more detailed data to improve prediction and clinical utility.

This study demonstrates that models can be produced using only data recorded in medical records and can attain predictive values of 70–80% to identify people at risk of developing AS in test data, although in the general population the positive predictive value would only be 0.15%-0.25% due to the low prevalence of the condition. This may be improved by incorporating the wider axial SpA spectrum which has a higher prevalence.

## Supporting information

**S1 Fig. Male participants with AS.**
(TIFF)

**S2 Fig. Female participants with AS.**
(TIFF)

**S1 Table. Principal component analysis results for males.**
(DOCX)

**S2 Table. Principal component analysis results for females.**
(DOCX)

**S3 Table. Code identified in both male and female analysis, male only or female only.**
(DOCX)

## Author Contributions

**Conceptualization:** Jonathan Kennedy, Natasha Kennedy, Ernest Choy, Stefan Siebert, Sinead Brophy.

**Data curation:** Jonathan Kennedy.

**Formal analysis:** Jonathan Kennedy, Natasha Kennedy, Roxanne Cooksey, Sinead Brophy.

**Investigation:** Jonathan Kennedy.

**Methodology:** Jonathan Kennedy, Natasha Kennedy.

**Supervision:** Sinead Brophy.

**Validation:** Jonathan Kennedy, Ernest Choy, Stefan Siebert, Sinead Brophy.

**Writing – original draft:** Jonathan Kennedy.

**Writing – review & editing:** Jonathan Kennedy, Natasha Kennedy, Roxanne Cooksey, Ernest Choy, Stefan Siebert, Muhammad Rahman, Sinead Brophy.

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
