## [Decision Letter · Decision Letter 0]

16 Aug 2021

PONE-D-20-36921

Predicting a diagnosis of ankylosing spondylitis using primary care health records – a machine learning approach.

PLOS ONE

Dear Dr. Kennedy,

Thank you for submitting your manuscript to PLOS ONE. After careful consideration, we feel that it has merit but does not fully meet PLOS ONE’s publication criteria as it currently stands. Therefore, we invite you to submit a revised version of the manuscript that addresses the points raised during the review process.

We look forward to receiving your revised manuscript.

Kind regards,

Alfredo Vellido

Academic Editor

PLOS ONE

Journal Requirements:

"This study was funded by UCB. The funders had no role in study design, data collection and analysis, decision to publish, or preparation of the manuscript. 

The SAIL databank was used and all data used in this study are available through the SAIL application process, https://www.saildatabank.com/application-process. 

This work was supported by Health Data Research UK , which is funded by the UK Medical Research Council, Engineering and Physical Sciences Research Council, Economic and Social Research Council, Department of Health and Social Care (England), Chief Scientist Office of the Scottish Government Health and Social Care Directorates, Health and Social Care Research and Development Division (Welsh Government), Public Health Agency (Northern Ireland), British Heart Foundation and the Wellcome Trust."

This work was supported by Health Data Research UK , which is funded by the UK Medical Research Council, Engineering and Physical Sciences Research Council, Economic and Social Research Council, Department of Health and Social Care (England), Chief Scientist Office of the Scottish Government Health and Social Care Directorates, Health and Social Care Research and Development Division (Welsh Government), Public Health Agency (Northern Ireland), British Heart Foundation and the Wellcome Trust."

"JK, SB reports grants from UCB Pharma, during the conduct of the study; and involved in (but not directly funded by) grants from Biogen, Sanofi and Novartis.

RC reports grants from Pfizer, during the conduct of the study; .

EC reports grants from UCB,  during the conduct of the study; grants and personal fees from Pfizer, grants and personal fees from UCB, grants from BioCancer, grants from Biogen, grants and personal fees from Novartis, grants and personal fees from Roche, personal fees from Amgen, personal fees from Chugai Pharma, personal fees from Eli Lilly, grants and personal fees from Sanofi, personal fees from Abbvie, personal fees from Janssen, personal fees from Gilead, personal fees from Bristol Myer Squibbs,  outside the submitted work;  In addition, Dr. Choy has a patent null pending.

SS reports grants and personal fees from AbbVie, grants and personal fees from UCB, grants and personal fees from Novartis, grants and personal fees from Janssen, grants and personal fees from Pfizer, grants from Bristol Myers Squibb,  from Boehringer-Ingelheim,  outside the submitted work."

6. PLOS requires an ORCID iD for the corresponding author in Editorial Manager on papers submitted after December 6th, 2016. Please ensure that you have an ORCID iD and that it is validated in Editorial Manager. To do this, go to ‘Update my Information’ (in the upper left-hand corner of the main menu), and click on the Fetch/Validate link next to the ORCID field. This will take you to the ORCID site and allow you to create a new iD or authenticate a pre-existing iD in Editorial Manager. Please see the following video for instructions on linking an ORCID iD to your Editorial Manager account: https://www.youtube.com/watch?v=_xcclfuvtxQ.

7. Please upload a new copy of Figure 1 and figure supplemental 1 & 2. as the detail is not clear. Please follow the link for more information: https://blogs.plos.org/plos/2019/06/looking-good-tips-for-creating-your-plos-figures-graphics/" https://blogs.plos.org/plos/2019/06/looking-good-tips-for-creating-your-plos-figures-graphics/.

Reviewers' comments:

Reviewer's Responses to Questions

**Comments to the Author**

1. Is the manuscript technically sound, and do the data support the conclusions?

Reviewer #1: Yes

Reviewer #2: Partly

2. Has the statistical analysis been performed appropriately and rigorously? 

Reviewer #1: Yes

Reviewer #2: Yes

3. Have the authors made all data underlying the findings in their manuscript fully available?

Reviewer #1: Yes

Reviewer #2: Yes

4. Is the manuscript presented in an intelligible fashion and written in standard English?

Reviewer #1: Yes

Reviewer #2: Yes

5. Review Comments to the Author

Reviewer #1: 1. Introduction section can be extended to add the issues in the context of the existing work

2. Literature review techniques have to be strengthened by including the issues in the current system and how the author proposes to overcome the same.

3. What is the motivation of the proposed work?

4. Research gaps, objectives of the proposed work should be clearly justified.

5. The authors should consider more recent research done in the field of their study (especially in the years 2018 and 2020 onwards). 6. The paper needs to provide significant experimental details to correctly assess its contribution: What is the validation procedure used?

7. Kindly provide several references to substantiate the claim made in the abstract (that is, provide references to other groups who do or have done research in this area).

8. An error and statistical analysis of data should be performed.

9. The conclusion should state scope for future work.

10. Discuss the future plans with respect to the research state of progress and its limitations.

11. Kindly refer the below paper:

1. Rajput, D.S., Basha, S.M., Xin, Q. et al. Providing diagnosis on diabetes using cloud computing environment to the people living in rural areas of India. J Ambient Intell Human Comput (2021). https://doi.org/10.1007/s12652-021-03154-4

Reviewer #2: *** METHODOLOGY ***

Novel machine learning methods can potentially identify

-> Are any of the ML methods employed truly novel or designed for the purposes of the experiment? If not, remove the word "novel".

In this study, 10% (n=543/5751) of patients were used to develop the model and the remaining patients with a diagnosis between the ages of 15 to 35 years (n=1559/5751, 27% of total) were used to examine the generalizability of the model. This means 63% of the data was not used in

the analysis as these patients were diagnosed with AS before the age of 15 or after the age of 35.

-> The 5751 seems to be from the identified male patients. What about the female? Or am I misunderstanding, and this should be made clearer?

This means 63% of the data was not used in the analysis as these patients were diagnosed with AS before the age of 15 or after the age of 35.

-> The majority was not used. NICE was mentioned as the reasoning behind this decision, but you should offer a further discussion as to the benefits vs. detriments of effectively discarding this much data.

Additionally, codes are aggregated so that concepts can be included in the features (e.g. a “pain” code flag may aggregate all the specific pain codes (pain in left thumb and pain in lumbar spine) into one combined “pain present” code

-> Unless I am misunderstanding, how is it beneficial to aggregate all the pain responses into 1 feature? I'd think it would make more sense to have a "pain in AS-associated areas" and "pain in other areas". (I think you actually do this as elaborated later, but then this sentence needs to be better worded.)

*** MODEL ***

79300 tests were run for all 793 AS patients and 100 controls each for the male and female cohorts

-> Shouldn't this be 555 (380+175), or were you also using the data set aside for generalizability? If you were, please explain why, because this would seem somewhat problematic.

The model selected is the one with the highest average F value across the 100 runs.

-> Why select only 1 model? May there be a benefit to using the previous 100 training models together as a random forest?

This list of significant variables was then transformed utilising principal component analysis to better understand the structure and relationship of the features to each other.

-> More information on PCA is needed in the manuscript body, e.g. how many principle components were kept after the transform and why was that number selected? Arbitrary?

*** RESULTS AND ANALYSIS ***

Table 1

-> Validation PPV and F-score are very low (0.15% and 0.30% respectively)?

Table 4

-> Validation PPV and F-score are very low (0.25% and 0.50% respectively)?

If we make the assumption that a “perfect” model would have 90% sensitivity and specificity as higher

19 percentages than this would likely be attributed to overfitting of the model.

-> One could counter overfitting e.g. by making better use of a validation set during hyperparameter tuning/optimization. (But this is a small point that you needn't necessarily do anything with, I'm just saying >90% should be achievable without overfitting.)

This implies that the female model is more accurate than the male model at identifying patients with AS.

-> Could it have something to do with the fact more principle components seemed to be used? Please explain.

It must be recognised that with low prevalence of AS in the general population, the positive predictive value in general and primary care clinical settings will always be very low.

-> I don't think this is necessarily true -- a model with extremely high sensitivity and specificity should be able to achieve a relatively high PPV. However, I acknowledge there are some limitations of your specific study -- and perhaps the nature of your model -- that inhibit your model's ability to get a high PPV. Perhaps this can be better clarified in the manuscript body (but it's not detrimental if it cannot).

The results of laboratory tests were not used in this model as they were too complex to interpret in the time available

-> Do you mean the project development time or the 3 years of time before diagnosis or something else? This should be specified.

Thus, 100 Chi-squared tests were run for each of the 3 years preceding the diagnosis date (700 chi-squared tests)

-> You should better explain why 700 tests were required (the reasoning behind the number).

There is some discussion mixed into the results, e.g. "This may imply that there is an increased reluctance to consider a diagnosis of AS or to perform spinal or pelvic x-rays for young female patients compared to their male counterparts" or "suggesting that women have a more complicated path to being diagnosed with AS, with multiple tests, diagnoses, and referrals, compared to men with AS" or "however, women are more likely to be given the test earlier than men. This is likely due to AS symptoms in females being suspected as a result of other conditions, such as RA" or "There was a smaller female cohort for analysis, which could have meant the study was underpowered to detect uveitis as a predictor in women." Please have a separate Discussion section for these and like comments.

*** WHAT YOU NEED YET INCLUDE ***

What software package did you use to implement your models? What were your hardware specs? What's the runtime of the model(s) on the system? What is the filesize of the models?

Needs more background and cited works, especially about other machine/deep-learning related AS-diagnostic works regardless of their inputs (e.g. images, features derived from images). Include at least:

-> http://dx.doi.org/10.1186/s42358-020-00126-8

-> https://doi.org/10.1016/j.compmedimag.2020.101718

-> https://doi.org/10.1007/s10067-019-04553-x

Please provide an ROC curve for your models and report AUC. If possible, provide some statistical analysis on the ROC curve (e.g. bootstrapped confidence intervals).

*** POSSIBLE DISCUSSION POINTS ***

This is likely due to the general population cohort having incomplete coverage of data compared to the other cohorts.

-> For generalizability, what is the likelihood that any given patient with back pain has in their record all the utilized data/stats? What could be done about it if they are missing some of the stats?

*** POSSIBLE RESEARCH EXPANSION ***

What about taking only the most common stats (that would be in the record of most patients) and possibly a shorter history time frame and re-running the experiment? How might results differ? It would be interesting to see and possibly include, but if you feel it is too out of scope consider it for future work.

*** GRAMMAR AND WORD CHOICE ***

AS is an important inflammatory musculoskeletal condition with effective therapy available,

especially if identified early

-> Word choice: "important"? Important how so? A better word could be used here to convey the meeting you want, which admittedly was somewhat vague to me.

especially for women with AS who have not well studied or recognised.

-> Grammar and meaning: "especially for women with AS, who traditionally have had later diagnoses possibly due to other factors; this area requires further research."

an unspecific back pain code present between age 15 and 20 AND a pelvic pain codes present between age 25 and 30.

-> Grammar: between ages X and Y (also elsewhere); a pelvic pain code or pelvic pain codes.

who hasan ESR test

-> Grammar: has an

The model was the applied to the unused individuals

-> Grammar: The model was applied to...

*** OTHER ***

Maybe combine the supplementary figures and write XXX individuals as XXX/YYY individuals, explaining that XXX denotes male and YYY denotes female.

The next stage would be to take the profiles of the decision trees forward so that we can identify people who fit the profile but are not diagnosed, and seek additional data about these patients from both medical records (e.g. results of laboratory tests such as elevated CRP) and to incorporate information directly from patients via an online patient reported measures questionnaire to capture more granularity and current symptoms (e.g. duration of pain, stiffness and pain worse in morning, pain improved with movement, buttock pain, hot burning joints, family history of SpA)

-> Sounds reasonable!

In the discussion, I really liked how you characterized the male and female "warning signs" and proposed actionable steps toward diagnosis in clinical practice.

I enjoyed reading the article and hope you use my suggestions to make it the best it can be! Have a good day.

6. PLOS authors have the option to publish the peer review history of their article (what does this mean?). If published, this will include your full peer review and any attached files.

Reviewer #1: No

Reviewer #2: No

---

## [Author Response · Author response to Decision Letter 0]

30 Jan 2022

Dear Editor,

The following is a response to the comments from the reviewers.

Reviewer #1: 

ResponseI feel I have responded to the reviewers points in the manuscript.

Reviewer #2: *** METHODOLOGY ***

Novel machine learning methods can potentially identify

-> Are any of the ML methods employed truly novel or designed for the purposes of the experiment? If not, remove the word "novel".

Response- The machine learning approaches are have been utilised in other industries (manufacturing, etc.) so are novel (new for this area) for this area. However, I have decided to remove this word to avoid confusion.

In this study, 10% (n=543/5751) of patients were used to develop the model and the remaining patients with a diagnosis between the ages of 15 to 35 years (n=1559/5751, 27% of total) were used to examine the generalizability of the model. This means 63% of the data was not used in

the analysis as these patients were diagnosed with AS before the age of 15 or after the age of 35.

-> The 5751 seems to be from the identified male patients. What about the female? Or am I misunderstanding, and this should be made clearer?

Response- That section does refer to male patients and I have edited the section to try and clarify the intention for clarity as suggested.

This means 63% of the data was not used in the analysis as these patients were diagnosed with AS before the age of 15 or after the age of 35.

-> The majority was not used. NICE was mentioned as the reasoning behind this decision, but you should offer a further discussion as to the benefits vs. detriments of effectively discarding this much data.

Response- Added to document.

Additionally, codes are aggregated so that concepts can be included in the features (e.g. a “pain” code flag may aggregate all the specific pain codes (pain in left thumb and pain in lumbar spine) into one combined “pain present” code

-> Unless I am misunderstanding, how is it beneficial to aggregate all the pain responses into 1 feature? I'd think it would make more sense to have a "pain in AS-associated areas" and "pain in other areas". (I think you actually do this as elaborated later, but then this sentence needs to be better worded.)

Response- Codes existed in their original state and in aggregated states. So, a ‘pain in back’ code would occur on its own and as part of a ‘pain’ code list. Additionally, the pain code list had a overall code list and was split into locations, e.g. a code for pain in lumbar spine would exist on its own, in a overall pain group and in a back pain group. The clinicians wanted to split the pain codes by locations rather than AS specific as male and female [patients can experience different locations of pain.

*** MODEL ***

79300 tests were run for all 793 AS patients and 100 controls each for the male and female cohorts

-> Shouldn't this be 555 (380+175), or were you also using the data set aside for generalizability? If you were, please explain why, because this would seem somewhat problematic.

Response-Your observation is correct. Adjusted in document.

The model selected is the one with the highest average F value across the 100 runs.

-> Why select only 1 model? May there be a benefit to using the previous 100 training models together as a random forest?

Response- It was decided to not employ a random forest approach as the output needed to be as understandable as possible to clinicians and patients to improve trust in the results.

This list of significant variables was then transformed utilising principal component analysis to better understand the structure and relationship of the features to each other.

-> More information on PCA is needed in the manuscript body, e.g. how many principle components were kept after the transform and why was that number selected? Arbitrary?

Response- The principle component were all kept and presented to the model, allowing the best significant variables to be selected.

*** RESULTS AND ANALYSIS ***

Table 1

-> Validation PPV and F-score are very low (0.15% and 0.30% respectively)?

Table 4

-> Validation PPV and F-score are very low (0.25% and 0.50% respectively)?

If we make the assumption that a “perfect” model would have 90% sensitivity and specificity as higher

19 percentages than this would likely be attributed to overfitting of the model.

-> One could counter overfitting e.g. by making better use of a validation set during hyperparameter tuning/optimization. (But this is a small point that you needn't necessarily do anything with, I'm just saying >90% should be achievable without overfitting.)

Response- Thank you for the comment I will keep it in mind for future work.

This implies that the female model is more accurate than the male model at identifying patients with AS.

-> Could it have something to do with the fact more principle components seemed to be used? Please explain.

Response- Added an additional comment.

It must be recognised that with low prevalence of AS in the general population, the positive predictive value in general and primary care clinical settings will always be very low.

-> I don't think this is necessarily true -- a model with extremely high sensitivity and specificity should be able to achieve a relatively high PPV. However, I acknowledge there are some limitations of your specific study -- and perhaps the nature of your model -- that inhibit your model's ability to get a high PPV. Perhaps this can be better clarified in the manuscript body (but it's not detrimental if it cannot).

The results of laboratory tests were not used in this model as they were too complex to interpret in the time available

-> Do you mean the project development time or the 3 years of time before diagnosis or something else? This should be specified.

Response- This refers to the routine data available. Since the project completed, SAIL has got access to lab results from the hospitals in Wales. However, a large amount of work is required to get this data in a suitable format for this kind of analysis.

Thus, 100 Chi-squared tests were run for each of the 3 years preceding the diagnosis date (700 chi-squared tests)

-> You should better explain why 700 tests were required (the reasoning behind the number).

Response- Tidied up in text.

There is some discussion mixed into the results, e.g. "This may imply that there is an increased reluctance to consider a diagnosis of AS or to perform spinal or pelvic x-rays for young female patients compared to their male counterparts" or "suggesting that women have a more complicated path to being diagnosed with AS, with multiple tests, diagnoses, and referrals, compared to men with AS" or "however, women are more likely to be given the test earlier than men. This is likely due to AS symptoms in females being suspected as a result of other conditions, such as RA" or "There was a smaller female cohort for analysis, which could have meant the study was underpowered to detect uveitis as a predictor in women." Please have a separate Discussion section for these and like comments.

*** WHAT YOU NEED YET INCLUDE ***

What software package did you use to implement your models? What were your hardware specs? What's the runtime of the model(s) on the system? What is the filesize of the models?

Response- The dataset was made employing SQL with data in SAIL. The model was run in R studio and had a runtime of < 24 hours for the complete run.

Needs more background and cited works, especially about other machine/deep-learning related AS-diagnostic works regardless of their inputs (e.g. images, features derived from images). Include at least:

-> http://dx.doi.org/10.1186/s42358-020-00126-8

-> https://doi.org/10.1016/j.compmedimag.2020.101718

-> https://doi.org/10.1007/s10067-019-04553-x

Please provide an ROC curve for your models and report AUC. If possible, provide some statistical analysis on the ROC curve (e.g. bootstrapped confidence intervals).

Response- Added to document.

*** POSSIBLE DISCUSSION POINTS ***

This is likely due to the general population cohort having incomplete coverage of data compared to the other cohorts.

-> For generalizability, what is the likelihood that any given patient with back pain has in their record all the utilized data/stats? What could be done about it if they are missing some of the stats?

Response- Many of these individuals were not selected due to failing to meet sufficient criteria for the main analysis. This means that many of these individuals may have incomplete history and relevant information may be missing from there timeline. In future it would be better to perform a validation on data from a different routine data source.

*** POSSIBLE RESEARCH EXPANSION ***

What about taking only the most common stats (that would be in the record of most patients) and possibly a shorter history time frame and re-running the experiment? How might results differ? It would be interesting to see and possibly include, but if you feel it is too out of scope consider it for future work.

Response- Having a shorter minimum history would increase the population size. 

*** GRAMMAR AND WORD CHOICE ***

AS is an important inflammatory musculoskeletal condition with effective therapy available,

especially if identified early

-> Word choice: "important"? Important how so? A better word could be used here to convey the meeting you want, which admittedly was somewhat vague to me.

Response- The word “important” has been removed to avoid confusion.

especially for women with AS who have not well studied or recognised.

-> Grammar and meaning: "especially for women with AS, who traditionally have had later diagnoses possibly due to other factors; this area requires further research."

Response- Changed in document.

an unspecific back pain code present between age 15 and 20 AND a pelvic pain codes present between age 25 and 30.

-> Grammar: between ages X and Y (also elsewhere); a pelvic pain code or pelvic pain codes.

Response- Changed in document.

who hasan ESR test

-> Grammar: has an

Response- The word “hasan” has been changed to “has an”.

The model was the applied to the unused individuals

-> Grammar: The model was applied to...

Response- Changed in document.

*** OTHER ***

Maybe combine the supplementary figures and write XXX individuals as XXX/YYY individuals, explaining that XXX denotes male and YYY denotes female.

Response- I think that even though this would make the supplementary figures more streamlined, it might make the figures more confusing to read.

The next stage would be to take the profiles of the decision trees forward so that we can identify people who fit the profile but are not diagnosed, and seek additional data about these patients from both medical records (e.g. results of laboratory tests such as elevated CRP) and to incorporate information directly from patients via an online patient reported measures questionnaire to capture more granularity and current symptoms (e.g. duration of pain, stiffness and pain worse in morning, pain improved with movement, buttock pain, hot burning joints, family history of SpA)

-> Sounds reasonable!

In the discussion, I really liked how you characterized the male and female "warning signs" and proposed actionable steps toward diagnosis in clinical practice.

I enjoyed reading the article and hope you use my suggestions to make it the best it can be! Have a good day.

Response- Thank you very much for your comments and suggestions. I feel they have made the document more clear.

Yours Faithfully,

Dr Jonathan Ian Kennedy.

---

## [Decision Letter · Decision Letter 1]

28 Feb 2022

PONE-D-20-36921R1Predicting a diagnosis of ankylosing spondylitis using primary care health records – a machine learning approach.PLOS ONE

Dear Dr. Kennedy,

Thank you for submitting your manuscript to PLOS ONE. After careful consideration, we feel that it has merit but does not fully meet PLOS ONE’s publication criteria as it currently stands. Therefore, we invite you to submit a revised version of the manuscript that addresses the points raised during the review process.

We look forward to receiving your revised manuscript.

Kind regards,

Alfredo Vellido

Academic Editor

PLOS ONE

Journal Requirements:

Additional Editor Comments (if provided):

Dear authors,

The revised version of your paper has improved on the original one, but you still need to provide a response to the fair further requests from reviewer #2.

Reviewers' comments:

Reviewer's Responses to Questions

**Comments to the Author**

1. If the authors have adequately addressed your comments raised in a previous round of review and you feel that this manuscript is now acceptable for publication, you may indicate that here to bypass the “Comments to the Author” section, enter your conflict of interest statement in the “Confidential to Editor” section, and submit your "Accept" recommendation.

Reviewer #2: (No Response)

2. Is the manuscript technically sound, and do the data support the conclusions?

Reviewer #2: Yes

3. Has the statistical analysis been performed appropriately and rigorously? 

Reviewer #2: No

4. Have the authors made all data underlying the findings in their manuscript fully available?

Reviewer #2: Yes

5. Is the manuscript presented in an intelligible fashion and written in standard English?

Reviewer #2: Yes

6. Review Comments to the Author

Reviewer #2: I've looked through the changes and the manuscript has certainly been improved. I can agree with most of the authors' modifications and/or refutations of my suggestions. Nevertheless, there are still a few things I would like clarified or included:

1. I still do not think it is clear why the Validation PPV and F-score are very low, compared to the train and test sets, in Tables 1 and 4. A more detailed description of how you used the validation set (e.g. vs the test set during model development/evaluation) may help.

2. Thanks for the information on model runtime (<24 hours) and development platform (R Studio). Please include these in the manuscript body. Furthermore, please include the system hardware specifications, and try to give a (more) specific runtime for the complete prediction/processing of 1 patient record through the model.

3. I would like to see in the manuscript a (bootstrapped) ROC curve over the final test data; please make and provide one. Since you are using R, the pROC package may be useful (https://www.rdocumentation.org/packages/pROC/versions/1.18.0). I have provided some sample (psuedo)code below to assist with this, where IN_DATA_TRUE and IN_DATA_PREDICT are vectors containing binary class info (e.g. 1 for AS and 0 for non-AS).

4. "This implies that the female model is more accurate than the male model at identifying patients with AS." -> In partial regard to this, you stated, "The principle component were all kept and presented to the model, allowing the best significant variables to be selected," which is a fair response. Could you provide some brief commentary on why you think the feature engineering process created more principle components for females than males, and to what degree (you believe) the number of components has on the final model accuracy?

Thanks for submitting your revised article, it was nice to read and exciting to see how it has progressed!

--- Sample pROC Code ---

library(pROC)

roc_obj <- roc(IN_DATA_TRUE, IN_DATA_PREDICT)

auc <- auc(roc_obj)

aucci <- ci.auc(roc_obj, boot.n=10000) # get 95% confidence interval

thresh <- ci.thresholds(roc_obj, boot.n=10000)

plot.new()

plot.roc(roc_obj, xlab="FPR", ylab="TPR", legacy=TRUE)

plot(thresh)

dev.off()

7. PLOS authors have the option to publish the peer review history of their article (what does this mean?). If published, this will include your full peer review and any attached files.

Reviewer #2: No

---

## [Author Response · Author response to Decision Letter 1]

21 Jun 2022

Thank you very much Reviewer 2 for recognising the improvements made between the two versions of this paper, we really appreciate it. Thank you for your further suggestions, we have endeavoured to make the changes or explanations required.

1. I still do not think it is clear why the Validation PPV and F-score are very low, compared to the train and test sets, in Tables 1 and 4. A more detailed description of how you used the validation set (e.g. vs the test set during model development/evaluation) may help.

The lower values observed in the validation dataset, in comparison to the training/testing datasets, can be explained in the following ways.

• The validation dataset includes records that were deemed not suitable for the main analysis, so those records might not have suitable data coverage and be of lower quality, resulting in incorrect classification.

• The models were developed on a 1:1 (case to control) basis and when presented with a less favourable ratio it became harder to achieve good PPV values as there is a much higher ratio of not cases in the general population compared to the training dataset. This is likely why many other models do not work well in the real world. However, the authors feel that the approach undertaken achieved some good hypotheses and results which can better aid the identification of AS. 

2. Thanks for the information on model runtime (<24 hours) and development platform (R Studio). Please include these in the manuscript body. Furthermore, please include the system hardware specifications, and try to give a (more) specific runtime for the complete prediction/processing of 1 patient record through the model.

Thank you for this feedback, the following section has been added to the paper (Materials and methods section: Dataset).

The dataset construction and analysis were undertaken in Eclipse and R Studio in the SAIL remote desktop server; the processor employed was an AMD EPYC 7543 32-Core Processor 2.79 GHz (2 processors) with 16.0 GB of RAM and running Windows 10. The run time for construction of the dataset and running the analysis was < 24 hours, with most of the time expended generating the dataset for the analysis to be undertaken. The processing of one patient took 32 seconds, with much of the time taken in R studio communicating with the SQL server.

3. I would like to see in the manuscript a (bootstrapped) ROC curve over the final test data; please make and provide one. Since you are using R, the pROC package may be useful (https://www.rdocumentation.org/packages/pROC/versions/1.18.0). I have provided some sample (psuedo)code below to assist with this, where IN_DATA_TRUE and IN_DATA_PREDICT are vectors containing binary class info (e.g. 1 for AS and 0 for non-AS).

This section requires graphs to answer which are included in the attachment.

4. "This implies that the female model is more accurate than the male model at identifying patients with AS." -> In partial regard to this, you stated, "The principle component were all kept and presented to the model, allowing the best significant variables to be selected," which is a fair response. Could you provide some brief commentary on why you think the feature engineering process created more principle components for females than males, and to what degree (you believe) the number of components has on the final model accuracy?

There are more cases of AS present in males than females. AS usually develops in young adults, and male/female have a different healthcare usage at this age. Young men tend to have a much lower usage than females so will likely have a simpler profile in their healthcare record than females. An additional reason might be that the symptoms expressed in a male would be identified as AS quicker, compared to females who might investigate other conditions before AS (e.g. RA, etc.).

---

## [Decision Letter · Decision Letter 2]

3 Aug 2022

PONE-D-20-36921R2

Predicting a diagnosis of ankylosing spondylitis using primary care health records – a machine learning approach.

PLOS ONE

Dear Dr. Kennedy,

Thank you for submitting your manuscript to PLOS ONE. After careful consideration, we feel that it has merit but does not fully meet PLOS ONE’s publication criteria as it currently stands. Therefore, we invite you to submit a revised version of the manuscript that addresses the points raised during the review process.

We look forward to receiving your revised manuscript.

Kind regards,

Alfredo Vellido

Academic Editor

PLOS ONE

Journal Requirements:

Reviewers' comments:

Reviewer's Responses to Questions

**Comments to the Author**

1. If the authors have adequately addressed your comments raised in a previous round of review and you feel that this manuscript is now acceptable for publication, you may indicate that here to bypass the “Comments to the Author” section, enter your conflict of interest statement in the “Confidential to Editor” section, and submit your "Accept" recommendation.

Reviewer #2: (No Response)

2. Is the manuscript technically sound, and do the data support the conclusions?

Reviewer #2: Yes

3. Has the statistical analysis been performed appropriately and rigorously? 

Reviewer #2: Yes

4. Have the authors made all data underlying the findings in their manuscript fully available?

Reviewer #2: Yes

5. Is the manuscript presented in an intelligible fashion and written in standard English?

Reviewer #2: Yes

6. Review Comments to the Author

Reviewer #2: All of my questions have been answered satisfactorily. Thank you very much!

The ROC curve and the corresponding ROC AUC must be reported in the manuscript body as part of the results. Furthermore, I believe the explanation about the validation set and the rationale for why there are more principle components for females should be incorporated into the manuscript as part of a discussion; this would make the manuscript stronger and give readers a better understanding of the results and implications.

Thereafter, this manuscript should be fit for publication.

7. PLOS authors have the option to publish the peer review history of their article (what does this mean?). If published, this will include your full peer review and any attached files.

Reviewer #2: No

---

## [Author Response · Author response to Decision Letter 2]

10 Oct 2022

Response to Reviewers

The ROC curves have been incorporated into the results section. The discussion section has been expanded to discuss why the male and female cohorts generate different amounts of principle components, as suggested by the reviewers.

The authors would like to thank that comment submitted by the reviewers for recognising the improvements made between the versions of this paper, we really appreciate it. Thank you for your further suggestions, we have endeavoured to make the changes or explanations required.

---

## [Decision Letter · Decision Letter 3]

1 Dec 2022

Predicting a diagnosis of ankylosing spondylitis using primary care health records – a machine learning approach.

PONE-D-20-36921R3

Dear Dr. Kennedy,

We’re pleased to inform you that your manuscript has been judged scientifically suitable for publication and will be formally accepted for publication once it meets all outstanding technical requirements.

Kind regards,

Alfredo Vellido

Academic Editor

PLOS ONE

Additional Editor Comments (optional):

Reviewers' comments:

Reviewer's Responses to Questions

**Comments to the Author**

1. If the authors have adequately addressed your comments raised in a previous round of review and you feel that this manuscript is now acceptable for publication, you may indicate that here to bypass the “Comments to the Author” section, enter your conflict of interest statement in the “Confidential to Editor” section, and submit your "Accept" recommendation.

Reviewer #2: All comments have been addressed

2. Is the manuscript technically sound, and do the data support the conclusions?

Reviewer #2: Yes

3. Has the statistical analysis been performed appropriately and rigorously? 

Reviewer #2: Yes

4. Have the authors made all data underlying the findings in their manuscript fully available?

Reviewer #2: Yes

5. Is the manuscript presented in an intelligible fashion and written in standard English?

Reviewer #2: Yes

6. Review Comments to the Author

Reviewer #2: I reviewed the part of the document containing the revision markings (pages 69 to 97 of the full PDF) and my concerns have all largely been addressed. I would just advise a proofread before publication (e.g. "an approach" rather than "a approach" where the original read "a new approach"). Additionally, "Limitations" should probably be its own (sub)section, rather than just "Limitations:" at the start of a paragraph.

Congratulations! I hope you all have a nice celebration and enjoy a well-deserved drink. :)

7. PLOS authors have the option to publish the peer review history of their article (what does this mean?). If published, this will include your full peer review and any attached files.

Reviewer #2: No

---

## [Editor Report · Acceptance letter]

24 Mar 2023

PONE-D-20-36921R3 

Predicting a diagnosis of ankylosing spondylitis using primary care health records – a machine learning approach. 

Dear Dr. Kennedy:

I'm pleased to inform you that your manuscript has been deemed suitable for publication in PLOS ONE. Congratulations! Your manuscript is now with our production department. 

Kind regards, 

on behalf of

Dr. Alfredo Vellido 

Academic Editor

PLOS ONE